# Visible light-mediated dearomative spirocyclization/imination of nonactivated arenes through energy transfer catalysis

Chao Zhou[1], Elena V. Stepanova[1,2], Andrey Shatskiy [1], Markus D. Kärkäs [1] & Peter Dinér [1] ✉

Aromatic compounds serve as key feedstocks in the chemical industry, typically undergoing functionalization or full reduction. However, partial reduction via dearomative sequences remains underexplored despite its potential to rapidly generate complex three-dimensional scaffolds and the existing dearomative strategies often require metal-mediated multistep processes or suffer from limited applicability. Herein, a photocatalytic radical cascade approach enabling dearomative difunctionalization through selective spirocyclization/imination of nonactivated arenes is reported. The method employs bifunctional oxime esters and carbonates to introduce multiple functional groups in a single step, forming spirocyclic motifs and iminyl functionalities via N−O bond cleavage, hydrogen-atom transfer, radical addition, spirocyclization, and radical-radical cross-coupling. The reaction constructs up to four bonds (C−O, C−C, C−N) from simple starting materials. Its broad applicability is demonstrated on various substrates, including pharmaceuticals, and it is compatible with scale-up under flow conditions, offering a streamlined approach to synthesizing highly decorated three-dimensional frameworks.

Aromatic compounds represent one of the most abundant chemical feedstocks and can be easily transformed into a wide range of functionalized two-dimensional scaffolds[1,2]. Conversely, their conversion into architecturally complex three-dimensional structures through dearomative processes[3–6] is intrinsically challenging due to the resonance stabilization of the aromatic systems[7]. The classical reduction reactions, such as catalytic hydrogenation[8] and Birch reduction[9], provide various dearomatized products, but without introducing additional functionalities into the ring system. Reactions that can simultaneously disrupt the aromatic system and introduce additional functionalities − dearomative difunctionalizations − has proved to be challenging. Complex difunctionalized products can be obtained with dearomative strategies via transition-metal−mediated activation, dearomative oxidation, and photocycloaddition manifolds[10–13]. Transition-metal−mediated dearomatizations have been widely used and enable elaborate functionalization schemes (Fig. 1, top left) through

coordination, electrophilic attack, nucleophilic attack, and decomplexation mechanism[14]. Dearomative oxidation manifolds include phenol oxidation[15] and radical-mediated oxidation[16–20] (Fig. 1, top left) that provide cyclohexadienone derivatives from the corresponding phenols or aryl ethers or microbial oxidations of monosubstituted arenes[21]; however, both of these reactions are hampered by limitations in the substrate scope. Dearomative photocycloaddition reactions have the ability to increase the structural complexity through the formation of bridged or fused bicyclic dearomatized products via [2 + 2], [3 + 2], and [4 + 2] cycloadditions[6,11,22–31], (Fig. 1, top right) and in some cases, the fused bicycles can undergo cycloreversion/fragmentation to yield dearomatized difunctionalized cyclic products (Fig. 1, top right). Although more sophisticated methods have increased the complexity of building blocks accessible from simple arenes[32], it is still pivotal to develop mild, selective, and versatile approaches for achieving dearomative difunctionalization of nonactivated arenes into

[1]Department of Chemistry, KTH Royal Institute of Technology, Stockholm, Sweden. [2]Research School of Chemistry & Applied Biomedical Sciences, Tomsk Polytechnic University, Tomsk, Russia. ✉e-mail: diner@kth.se

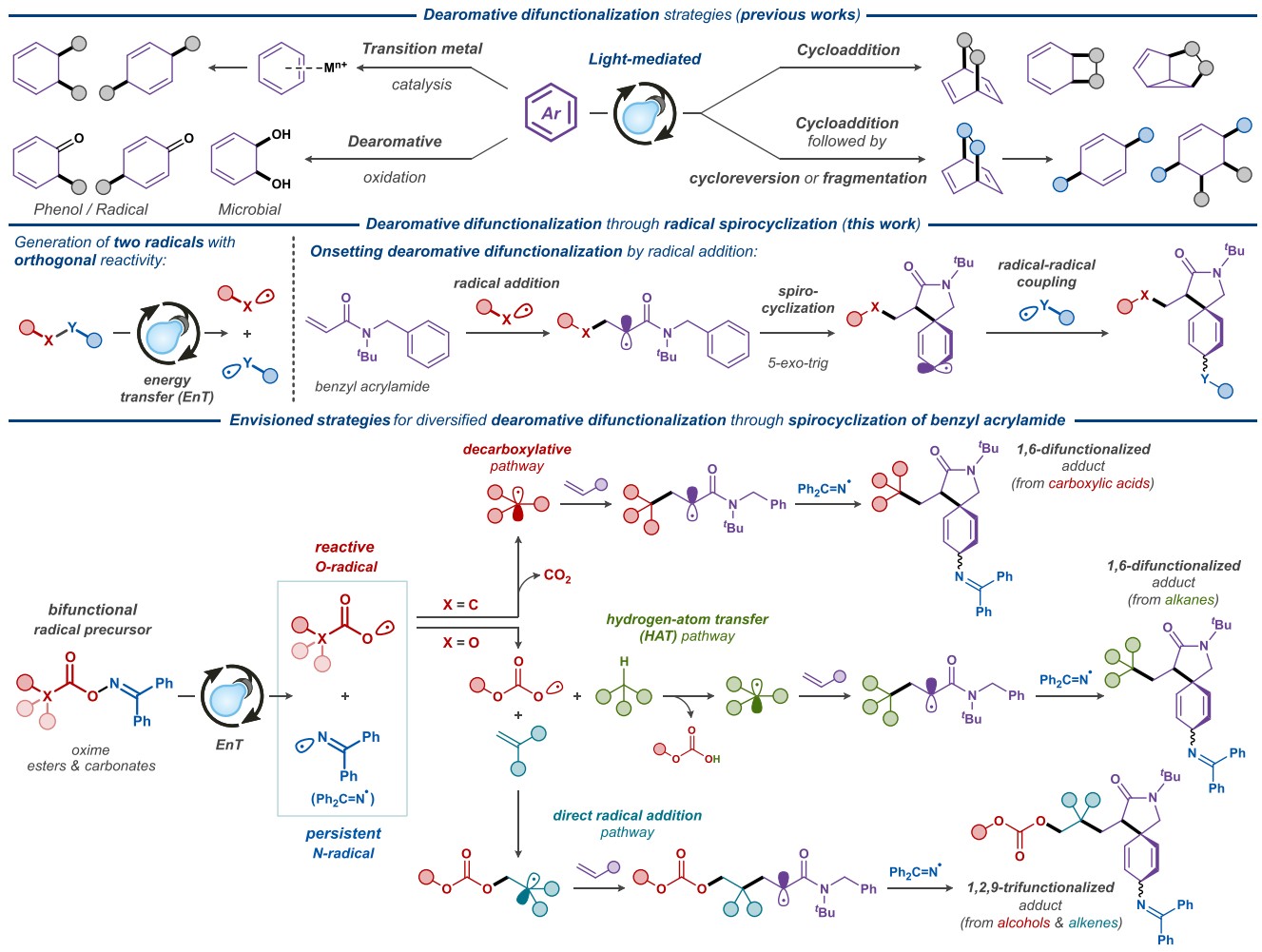

**Fig. 1 | Dearomative strategies for partial reduction of aromatic motifs.** Top: Established strategies for dearomative difunctionalization. Middle: Envisioned reaction design for the photocatalytic dearomative difunctionalizations via spirocyclic iminations. Bottom: Mechanistic pathways for the proposed dearomative difunctionalizations.

architecturally more complex three-dimensional structures via multicomponent reactions.

## Results

### Reaction design

Recently, we disclosed an effective photoredox-mediated dearomative annulation approach to spirocyclic compounds through homolytic C−O bond activation of aromatic carboxylic acids[33]. The key spirocyclization step was realized through 6-*exo*-trig or 5-*exo*-trig intramolecular C-radical addition to a benzene ring, producing a cyclohexadienyl C-radical species, which is converted to the final 1,4-hexadienyl product upon one-electron/one-proton reduction. Despite numerous attempts to form difunctionalized dearomatization products, we were unsuccessful in trapping the cyclohexadienyl C-radical intermediate with different somophiles to form C−C or C−heteroatom bonds. This is in line with the known propensity of this C-radical to undergo either one-electron reduction[34,35] or oxidation[16,20] followed by a bond-forming reaction with nucleophilic or electrophilic reagents.

Therefore, we sought to circumvent this limitation using the persistent radical effect[36,37], which involves trapping of the transient cyclohexadienyl C-radical with a persistent radical generated in situ (Fig. 1, middle). The envisioned strategy is initiated by homolytic bond cleavage in a radical precursor to produce both reactive and persistent radicals, the first of which onsets spirocyclization of the radical

acceptor, while the second is utilized for functionalization of the spirocyclic cyclohexadienyl C-radical through radical-radical coupling. Recently, O-functionalized oximes has emerged as a versatile class of bifunctional radical precursors, which can be harnessed to produce a transient O-radical and a persistent iminyl N-radical species upon photocatalytic energy transfer[38,39]. Typically, these systems employ ubiquitous iridium-based photocatalysts, such as [Ir(dF(CF$_3$) ppy)$_2$(dtbbpy)](PF$_6$), which is excited under visible light irradiation and mediates the intermolecular energy transfer to the radical precursor, onsetting the homolytic N−O bond cleavage process[40]. Inspired by the pioneering works by Glorius[41–43], Molander[44], and others[45–49], we envisioned three different approaches to 1,4-difunctionalization of nonactivated arenes concomitant with dearomative spirocyclization using various oxime radical precursors in conjunction with secondary coupling partners (Fig. 1, bottom).

In the first of the envisioned systems, oxime esters of carboxylic acids are employed as progenitors to a carboxylate O-radical species, which undergoes facile decarboxylation to afford a nucleophilic C-radical (Fig. 1, bottom, decarboxylation pathway). The latter readily adds to the electron-deficient benzyl acrylamide acceptor to promote the onset of the spirocyclization reaction, followed by coupling of the produced cyclohexadienyl C-radical with the persistent iminyl N-radical to furnish the desired difunctionalized product. Alternatively, the employment of oxime carbonates as the bifunctional

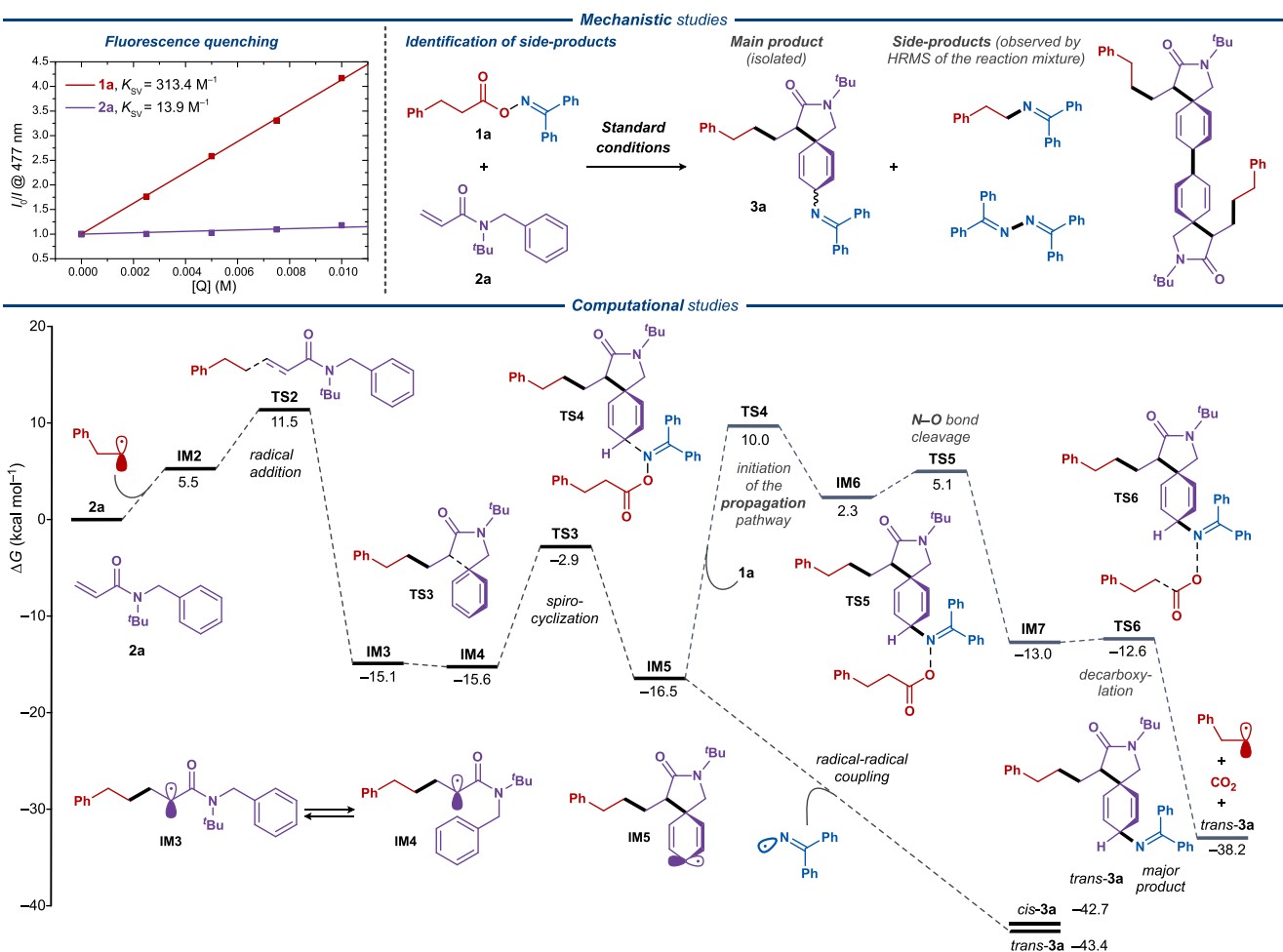

**Fig. 2 | Mechanistic studies of the dearomative spirocyclic iminations.** Top left: Stern-Volmer fluorescent quenching study with [Ir(dF(CF₃)ppy)₂(dtbbpy)](PF₆) photocatalyst (15 μM) and compounds **1a** and **2a** as quenchers. Top right: The main product and side products identified in the reaction mixture by LC-HRMS analysis. Bottom: Gibbs free energy diagram for the proposed transformation optimized at the B3LYP/6-311 + G(d,p) level using the Grimme correction for dispersion (D3) and the Conductor-like Polarizable Continuum Model (CPCM, UFF, ethyl acetate). Gibbs free energies are given in kcal mol⁻¹.

radical precursors provides a more kinetically stable carbonate O-radical species, which is less prone to decarboxylation[50] and does not undergo direct addition to the electron-deficient benzyl acrylamide acceptor due to the polarity mismatch. In this second envisioned system, the reactivity of such carboxylate O-radical is redirected by the addition to an electron-rich alkene (Fig. 1, bottom, direct radical addition pathway), and the produced C-radical species then onsets the radical addition/spirocyclization/imination sequence leading to the desired product. In the third system, the carbonate O-radical[51] is instead utilized as a hydrogen-atom transfer (HAT) agent, allowing generation of the key nucleophilic C-radical species from C−H bonds in both activated and aliphatic substrates (Fig. 1, bottom, HAT pathway). Thereby, our outlined photocatalytic systems provide three alternative routes to the synthesis of highly functionalized spirocyclic lactams decorated with an iminyl group through dearomatization/radical coupling cascade reactions.

To assess the feasibility of the envisioned photocatalytic systems, we investigated the reactivity of the proposed key radical intermediates using density functional theory (DFT) calculations (for details, see Supplementary Information). The envisioned reaction sequence is initiated by excitation of the model photocatalyst [Ir(dF(CF₃)ppy)₂(dtbbpy)]⁺ to the triplet excited state (57.8 kcal mol⁻¹). The latter is quenched by the diphenyl oxime ester **1a** through energy transfer to generate the triplet excited state oxime ester **1a\*** (44.3 kcal mol⁻¹)[38],

which undergoes low barrier fragmentation (2.3 kcal mol⁻¹) to carbon dioxide, an N-centered diphenyl iminyl radical and a primary alkyl C-radical (Supplementary Fig. S9). The feasibility of the designed mechanistic pathway relies on the difference in reactivities of the formed N- and C-centered radicals, manifested in the difference in Gibbs free energy for their homo- and cross-coupling reactions (N−N: −34.3 kcal mol⁻¹, C−N: −55 kcal mol⁻¹, C−C: −69.9 kcal mol⁻¹). Here, despite the strong driving force for the radical−radical coupling reactions, their rate is greatly limited due to the low steady-state concentration of the coupling components, enabling the trapping of the free radical species with a suitable somophile and onsetting a dearomative radical cascade pathway. According to the DFT calculations, C-centered radicals (aliphatic, acyl, and oxyacyl) display higher reactivity (ΔG‡ = 11.6, 10.4, and 11.8 kcal mol⁻¹, for the aliphatic, acyl, and oxyacyl radicals, respectively, Supplementary Fig. S10) towards the electron-deficient somophile **2a** compared to the N-centered iminyl radical (ΔG‡ = 18.0 kcal mol⁻¹), corroborating the feasibility of the proposed transformation.

The reaction conditions for the first of the envisioned systems were optimized with oxime **1a** and acrylamide **2a** as the model substrates, leading to the difunctionalized spirocyclic product **3a**. As the result of the optimization study (for additional details, see the Supplementary Information and Supplementary Table S1–S5), imination product **3a** was obtained in 56% yield and 1.4:1 dr in ethyl acetate and

with [Ir(dF(CF$_3$)ppy)$_2$(dtbbpy)](PF$_6$) as the photocatalyst under visible light irradiation (440 nm LED) (Supplementary Table S1). Control experiments confirmed the necessity of both the light and the iridium photocatalyst for the reaction under 440 nm irradiation, and direct excitation at shorter wavelengths (390 nm) gave significantly lower yields. The proposed mechanistic sequence was further supported by fluorescence quenching studies, where only the oxime radical precursor **1a** was found to effectively quench the excited iridium photocatalyst (Fig. 2, top), while no appreciable quenching was observed for alkene **2a** (see Fig. 2 and Supplementary Fig. S1–S2).

Analysis of the crude reaction mixture under standard conditions between **1a** and **2a** by LC-HRMS demonstrated not only the formation of the desired product **3a**, but also several side-products consistent with the proposed mechanistic sequence. These side-products include the radical-radical coupling adducts between the proposed intermediates, including one C−N cross-coupling adduct between the iminyl radical and the aliphatic radical, as well as two homodimerization adducts (Fig. 2, top right; Supplementary Fig. S6). Additional trapping experiments in the presence of the radical scavenger TEMPO (2 equiv.) under otherwise standard conditions completely inhibited the formation of product **3a** and led to the formation of the coupling product between TEMPO and the C-centered aliphatic radical as detected by HRMS (Supplementary Fig. S6).

The DFT calculations suggest that upon homolytic N−O-bond cleavage, the C-centered radical initiates the radical-cascade process through radical addition to somophile **2a** via **TS2** ($\Delta G^{\ddagger} = 11.5$ kcal mol$^{-1}$), forming an α-carbonyl radical in an exothermic process (**IM3**, $\Delta G = -15.1$ kcal mol$^{-1}$) (Fig. 2, bottom). The α-carbonyl radical engages in a rate-limiting 5-*exo*-trig spirocyclization via **TS3** ($\Delta G^{\ddagger} = 12.7$ kcal mol$^{-1}$) to yield intermediate **IM5**, which subsequently engages in hetero-coupling with the iminyl radical to provide the final imination product as a mixture of diastereomers (*trans*-**3a** and *cis*-**3a**) with the *trans*-diastereomer being slightly more thermodynamically favored. The DFT calculations suggest a considerably high reaction barrier for the potential propagation step (**TS4**, $\Delta G^{\ddagger} = 26.5$ kcal mol$^{-1}$), suggesting that the envisioned reaction does not proceed through the radical chain mechanism, which is further supported by light on-off experiments (Supplementary Fig. S7).

## Investigation of the substrate scope

With the optimized reaction conditions in hand, the generality of the photocatalytic decarboxylative dearomative difunctionalization was evaluated with a variety of oxime esters (Fig. 3). Primary alkyl C-radicals were smoothly incorporated into the dearomatized products with moderate yields (**3a**–**3b**, 54–60%). Remarkably, the methyl radical produced the desired product in good yield (**3c**, 46%), which is generally challenging due to its high reactivity.

Similarly, secondary C-radicals, including cyclobutyl, cyclohexyl, and *gem*-difluorinated cyclohexyl radicals, were well-tolerated, affording the expected imination products in moderate yields (**3d**–**3f**, 56–61%). Furthermore, substrates equipped with bulkier tertiary butyl and adamantyl groups provided the expected products in slightly lower yields (**3g**–**3h**, 44–50%). Next, we wanted to investigate the spiroiminative reactivity of the aromatic moiety. It should be noted that the obtained compounds from the unsymmetrically substituted aromatic systems have 4 possible diastereomeric pairs arising from the three stereogenic centers of the product, and thus, the target compounds were either isolated as individual compounds or as a mixture of stereoisomers (see Fig. 3). Gratifyingly, replacing the phenyl ring with aromatic heteroatom containing heterocycles, such as thiophene and dibenzofuran, both gave the expected products in satisfying yield (**3i**: 52% and **3j**: 71%, respectively). The substrate with a fluoro-substituent in the 2-position (**3k**) of the aromatic ring increased the yield to 73%. A phenyl substituent was tolerated in the *para*-position of the spirocyclic motif, which gave rise to highly substituted compounds

(**3l**) in good yield (33%), considering the complexity of the synthesized products.

Apart from alkyl radicals, acyl and oxyacyl radicals were also suitable coupling partners to yield the corresponding spirocyclic derivatives (**3m**–**3p**, 57–63%). Next, we turned our attention to the exploration of functionalized acrylamide acceptors. Several symmetrical disubstituted substrates bearing methoxy, methyl, and chloro groups at the aromatic functionality furnished the desired products in reasonable yields (**3q**–**3t**, 39–58%). Other modifications of the amide functionality in the somophile, such as the substitution of the methylene group with a carbonyl functionality, provided the corresponding spiro-succinimide product in similar yields (**3u**, 43%, 3:1 dr). Substitution of the *N*-*t*Bu group with *N*-*i*Pr led to a decrease in yield (**3v**, 29%), while having secondary amide gave no product formation. Previously, the importance of the *t*-butyl substituent on the amide functionality was attributed to the decreased spatial distance between the two carbons that undergo the spirocyclization[35], but it has been shown that the *tert*-butyl group can be easily removed under mild conditions using copper(II) triflate[52].

To investigate the compatibility of the developed protocol for late-stage functionalization of bioactive molecules, several pharmaceutical drugs, amino acid derivatives, and peptides were converted to oxime esters and subjected to our decarboxylative protocol. Gemfibrozil, Fenofibric acid, and Ciprofibric acid, that are used in the treatment of abnormal blood lipid levels, delivered the expected spirocyclic dearomatized products in reasonable yields (50%, 40%, and 38%, for **3w**, **3x**, and **3y**, respectively). Other drugs, such as the nonsteroidal anti-inflammatory drug Oxaprozin and the strained β-lactamase inhibitor Sulfbactam, were compatible with the developed protocol and gave the corresponding spirocyclic compounds in moderate yields (**3z**: 40% and **3aa**: 19%, respectively). Further, the peptide-based Nateglinide (used in diabetes treatment), tosyl-protected glycine, and a sugar-based diprogulic acid were also compatible under these conditions (**3ab**: 33%, **3ac**: 49%, and **3ad**: 38%).

The transformation was also applied to the malonate oxime ester **1ae** that directly generates the α-carbonyl ester without a Giese type coupling reaction which gave the unsubstituted dearomatized spirocyclic/imination product **3ae** in 41% yield.

To demonstrate the versatility of the developed protocol, we wanted to expand the dearomative difunctionalization via a three-component system that utilizes our dearomative spirocyclization/imination of nonactivated arenes (Fig. 1, bottom) in combination with electron-rich and electron-poor alkenes. In this photocatalytic system, the radicals are generated from oxime carbonates via energy transfer from the photocatalyst and reacts through a radical cascade with both a more electron-rich and an electron-poor alkene. This dearomative spirocyclization proceed through five different types of radical intermediates and gives rise to the formation of four chemical bonds (C−O, C−C, C−C, and C−N). The fluorescence quenching studies for this catalytic system demonstrated that only the oxime carbonate **4a** effectively quenches the excited iridium photocatalyst (Fig. 2, top), while no appreciable quenching was observed for the electron-rich alkene **5a** (see Supplementary Figs. S3, S4, and S5). In addition, the DFT calculations revealed that the electron-deficient O-centered carbonate radical preferentially adds to the electron-rich alkene **5a** ($\Delta G^{\ddagger} = 7.8$ kcal mol$^{-1}$) rather than to the more electron-poor benzyl acrylate-based alkene **2a** ($\Delta G^{\ddagger} = 9.6$ kcal mol$^{-1}$) (Supplementary Fig. S11), which is in line with the control experiment where **4a** rather reacted with **5a** than with **2a** to provide the 1,2-oxyimination product (Supplementary Fig. S8a and S8b).

The formed C-centered radical displayed opposite selectivity, preferentially adding to the electron-deficient benzyl acrylate alkene **2a** ($\Delta G^{\ddagger} = 12.1$ kcal mol$^{-1}$) rather than to the more electron-rich alkene **5a** ($\Delta G^{\ddagger} = 16.1$ kcal mol$^{-1}$) (Supplementary Fig. S11). Thus, the DFT calculations indicate that the envisioned three-component reaction cascade is indeed feasible.

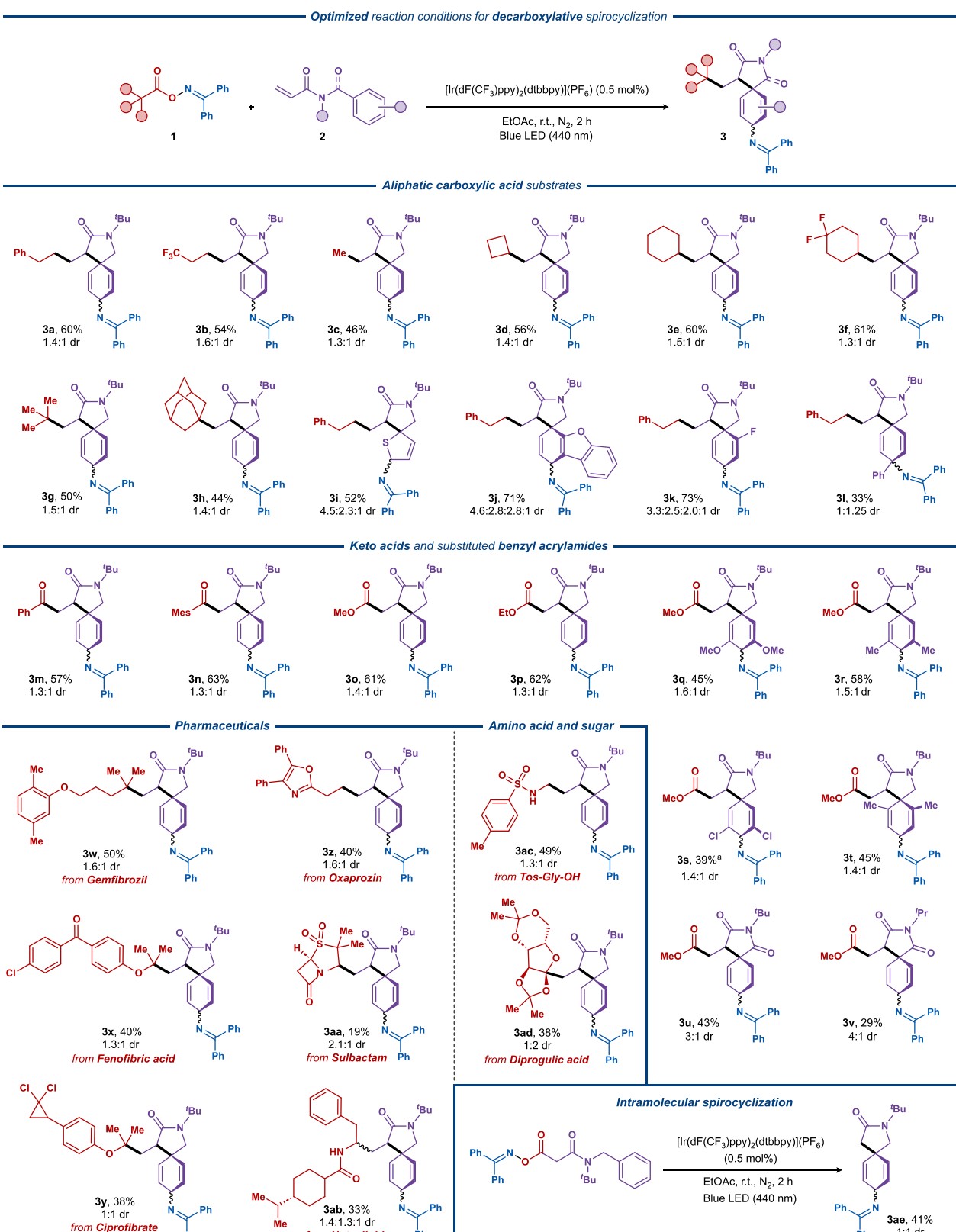

**Fig. 3 | The scope of the light-mediated decarboxylative dearomative difunctionalization.** Reaction conditions: oxime **1** (0.3 mmol, 1.5 equiv.), acrylamide **2** (0.2 mmol, 1.0 equiv.), [Ir(dF(CF₃)ppy)₂(dtbbpy)](PF₆) (0.5 mol%), EtOAc (3 mL), N₂, blue LEDs (440 nm, 10 W), 2 h, fan cooling (35–40 °C). Diastereomeric ratios (dr) were determined by ¹H NMR of the crude reaction mixtures. ᵃ4 h reaction time.

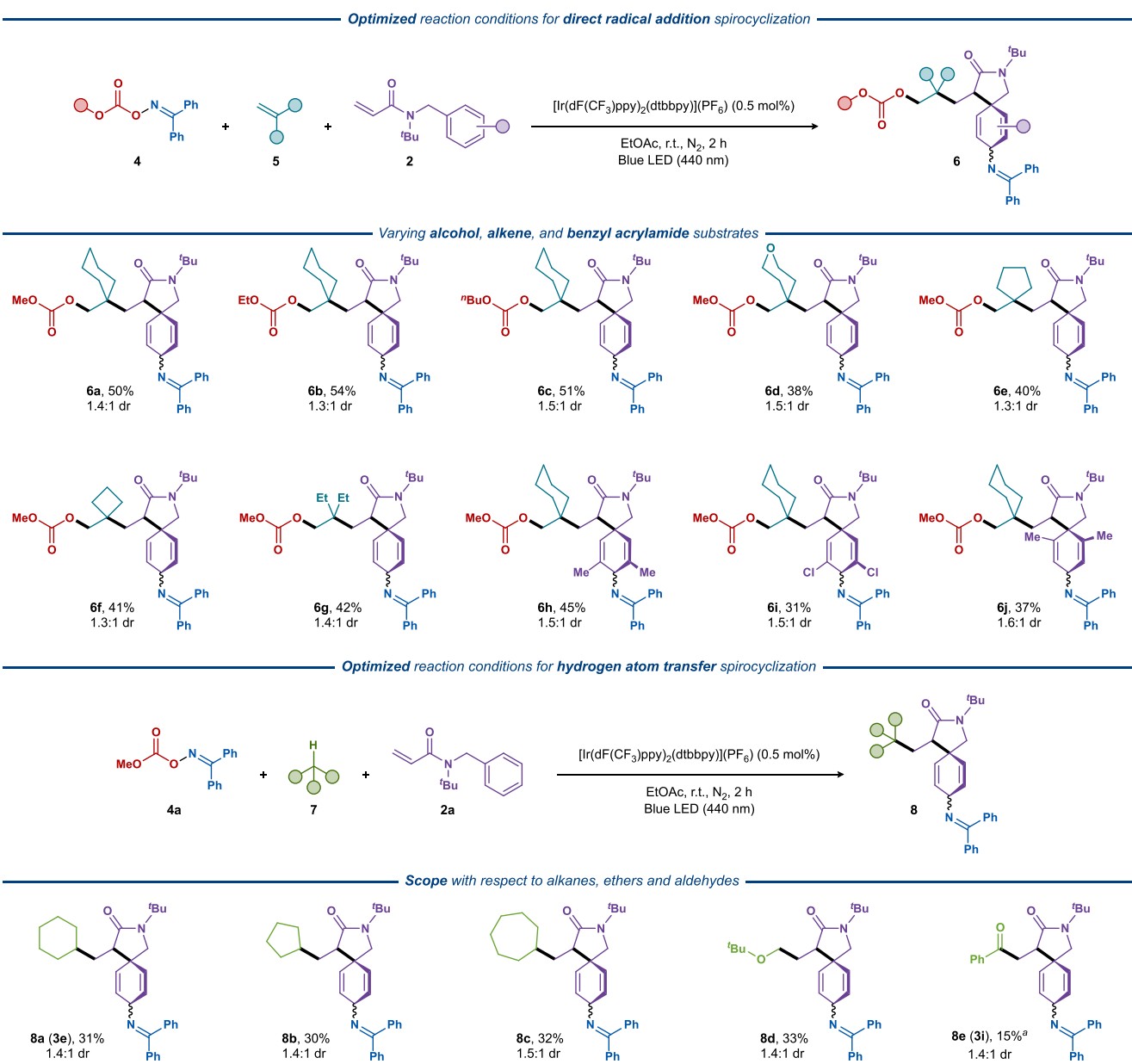

**Fig. 4 | Three-component direct radical addition–dearomative difunctionalization through spirocyclization.** Reaction conditions: oxime carbonate **4** (0.3 mmol, 1.5 equiv.), acrylamide **2** (0.2 mmol, 1.0 equiv.), non-activated alkene **5** (0.4 mmol, 2.0 equiv.), [Ir(dF(CF$_3$)ppy)$_2$(dtbbpy)](PF$_6$) (0.5 mol%), EtOAc (3 mL), N$_2$, blue LEDs (440 nm, 10 W), 4 h, fan cooling (35–40 °C). Diastereomeric ratios were determined by $^1$H NMR spectroscopy. Bottom: Scope of the hydrogen atom transfer–dearomative difunctionalization through spirocyclization. Reaction conditions: oxime carbonate **4** (0.4 mmol, 1.5 equiv.), acrylamide **2** (0.2 mmol, 1.0 equiv.), aliphatic substrate **7** (2 mL), [Ir(dF(CF$_3$)ppy)$_2$(dtbbpy)](PF$_6$) (0.5 mol%), EtOAc (1 mL), N$_2$, blue LEDs (440 nm, 10 W), 4 h, fan cooling (35–40 °C). Diastereomeric ratios were determined by $^1$H NMR spectroscopy. $^a$ benzaldehyde (1 mL), EtOAc (2 mL).

The scope of the direct radical addition–dearomative difunctionalization manifold was initially evaluated with various oxime carbonates using the same standard conditions as in the previous photocatalytic system but with 2 equiv. of the electron-rich alkene somophile (Fig. 4, top). Oxime carbonates derived from methyl, ethyl, and *n*-butyl alcohols all provided the corresponding products in almost the same yields (**6a**–**6c**, 50–54%) as the two-component system. A range of symmetrical 1,1-disubstituted nonactivated olefins smoothly reacted with **5a** and **2a** to deliver products **6d**–**6g** in similar yields (38–42%). Finally, acrylamide acceptors bearing methyl and chloro substituents in the aromatic ring were also well-tolerated (**6h**–**6j**, 31–45%).

The O-centered carbonate radicals exhibit a versatile reactivity profile that includes decarboxylation and addition to alkenes or aromatic systems[51,53,54]. Similar to other high-energy O-centered radicals, such as oxyl[55,56], oxyacyl[57,58], and phosphate[59,60] radicals, the O-centered carbonate radicals are potent HAT agents. Hence, unactivated alkanes can be exploited as the source of C-centered radicals that can engage in numerous reactions with somophiles. The reactivity of the carbonate radical was tested in several control experiments. In the absence of an electron-rich alkene, no spirocyclic-imination product was formed suggesting a competing reactivity for the carbonate radical (Supplementary Fig. S8a). On the other hand, performing a similar reaction in the absence of an electron-poor alkene and in the

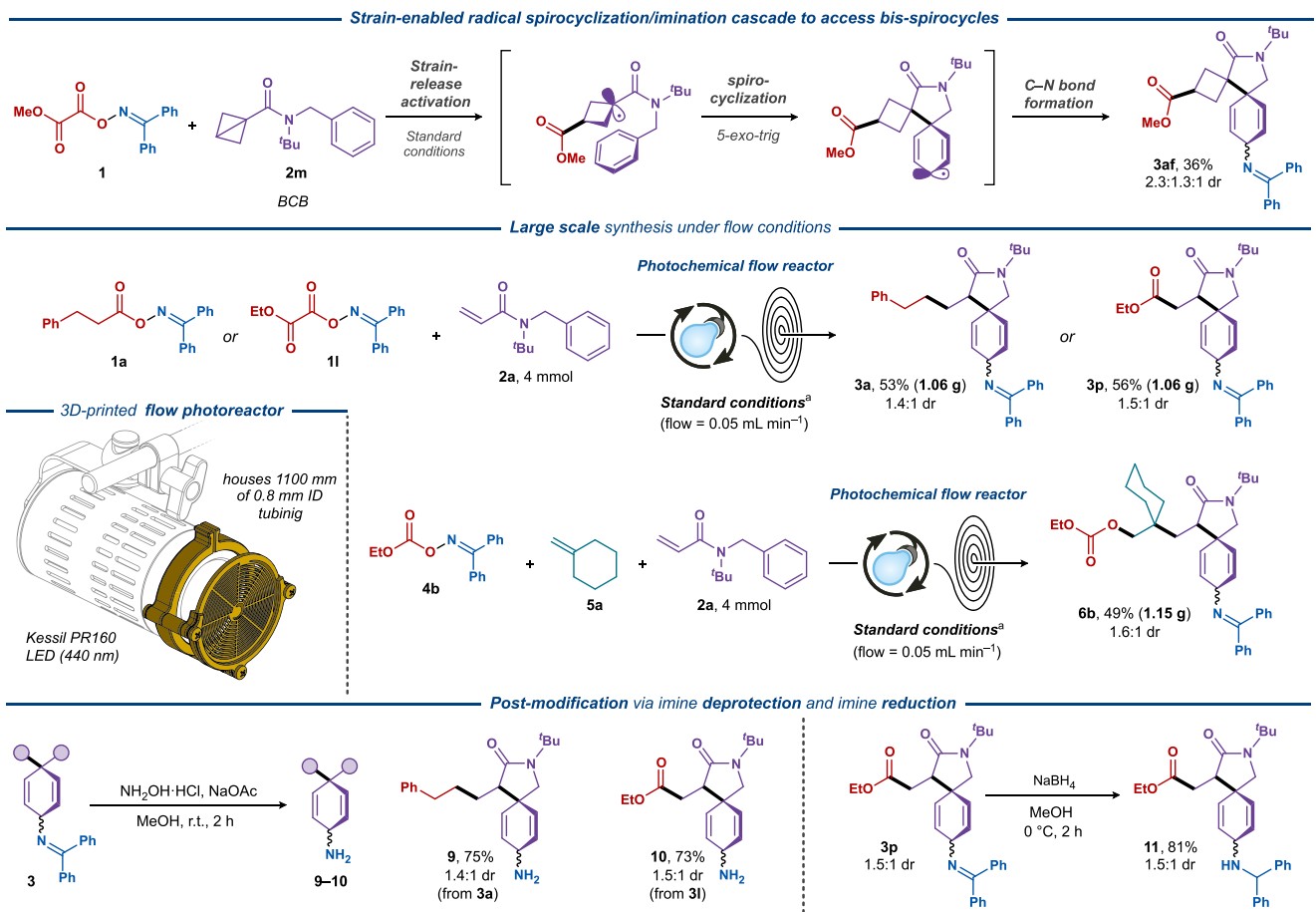

**Fig. 5 | Synthetic applications for dearomative spirocyclic/imination cascades.** Top: BCB-enabled formation of *bis*-spirocycles. Middle: Large-scale decarboxylative dearomative difunctionalization under continuous flow conditions. Bottom left: Deprotection of the imine functionality for selected products. Bottom right: Reduction of the imine functionality.

presence of an electron-rich alkene yields the difunctionalized product (Supplementary Fig. S8b) as previous reported by Molander and Glorius[41–44]. Upon irradiation of **4a**, the imination product of cyclohexane was detected by HRMS, which shows the potential HAT reactivity of the carbonate radical. This was further supported by DFT calculations that suggest that HAT from cyclohexane ($\Delta G^{\ddagger} = 7.8$ kcal mol$^{-1}$) is several orders of magnitude faster than HAT from ethyl acetate ($\Delta G^{\ddagger} = 10.0$ kcal mol$^{-1}$) that is used as the solvent under the developed conditions (Supplementary Fig. S12). The carbonate radical has only a slightly higher barrier for the competing decarboxylation reaction (9.3 kcal mol$^{-1}$) compared to HAT from cyclohexane; however, the use of a large excess of cyclohexane is expected to favor the desired HAT reaction (Supplementary Fig. S12). Thus, the dearomative difunctionalization manifold discussed above was further extended to employ transient carbonate radical species as potent HAT agents (Fig. 4, bottom). In this photocatalytic system, non-activated alkanes engaged in a competitive HAT reaction, onsetting the formation of the key C-radical species. Several alkane solvents, such as cyclohexane, cyclopentane, and cycloheptane, were easily incorporated into the spirocyclic frameworks without prefunctionalization (**8a–8c**, 30–32%). In addition, the α-C–H bond of ether and even the C(O)–H bond of an aldehyde was selectively activated to furnish the dearomative difunctionalization products, albeit in lower yields (**8d–8e**, 15–33%).

To further demonstrate the synthetic versatility of the developed methodology, we sought to utilize the *N-tert*-butyl-*N*-benzyl motif to access bis-spirocyclic products via a radical cascade process involving

energy transfer, strain-release activation, spirocyclization and C–N-bond forming imination. Recently, strain-release transformations of bicyclo[1.1.0]butanes (BCB) have facilitated the formation of reactive radical intermediates via photocatalysis to yield molecular frameworks in organic synthesis through the release of substantial ring strain energy[31,61–68].

To our delight, the utilization of the BCB-benzylamide (**2m**) successfully yielded a tricyclic product with two spirocyclic motif in a single transformation (**3af**, 36%).

The synthetic utility was further demonstrated through scale-up reactions using flow chemistry with an in-house designed flow reactor (Fig. 5). The reaction could be easily scaled up from 0.2 mmol to 4 mmol scale under flow conditions with nearly identical yields compared to the small-scale batch reactions, providing the expected products **3a**, **3p**, and **6b** in 53%, 56%, and 49% yields, respectively. Further synthetic manipulations of the dearomative difunctionalization products included the efficient conversion of the iminyl groups to the corresponding cyclohexadienyl amines in the presence of hydroxylamine (**9–10**, 73–75%)[69] and the reduction to the corresponding amine (**11**, 81%) using NaBH$_4$, which highlights their expedient access to both primary and secondary amines from the spirocyclic compounds.

## Discussion

In conclusion, the present work describes a groundbreaking photocatalytic methodology enabling dearomative difunctionalization of nonactivated arenes. The utilization of bifunctional oxime esters and

carbonates enabled single-step installation of both spiro- and iminyl-functionalities with large substrate scopes and functional group compatibility. The radical cascade involves N−O bond cleavage, $CO_2$ release, radical addition, 5-*exo*-trig cyclization, and radical cross-coupling, to form complex spiro-iminyl architectures. This method stands out for its versatility, accommodating a wide substrate scope and functional group compatibility, and its ability to establish up to four chemical bonds (C−O, 2 x C−C, and C−N) in a single synthetic step. Oxime carbonates were also utilized as hydrogen-atom transfer agents to activate C-H bonds from unactivated hydrocarbon solvents in the dearomative spirocyclic/imination and thereby enhancing the protocol's utility. Several late-stage functionalizations of bioactive molecules and the use of strain-release activation further exemplifies the procedure's practical relevance and adaptability. This work represents a significant advancement in photocatalytic chemistry, providing a versatile and selective tool for accessing high-value molecular architectures.

## Methods

### General procedure A for the two-component dearomative spirocyclization/imination of nonactivated arenes via photocatalytic radical relay

A 10 mL vial equipped with a magnetic stir bar was charged with oxime **1a** (0.3 mmol), *N*-benzyl-*N*-(*tert*-butyl)acrylamide **2a** (0.2 mmol), [Ir(dF(CF$_3$)ppy)$_2$(dtbbpy)](PF$_6$) (0.001 mmol) in EtOAc (3 mL). After the degassing with N$_2$ for 10 min, the mixture was irradiated by blue LEDs ($\lambda$ = 440 nm, 10 W) for 2 h at room temperature. After irradiation, the resulting homogenous solution was transferred to a 25 mL round bottom flask with the aid of CH$_2$Cl$_2$ (2 × 3 mL). NEt$_3$ (approx. 0.5 mL) and SiO$_2$ were added to this solution, and the volatiles were removed under reduced pressure, affording a powder which was loaded on the column. Purification by column chromatography using pre-basified silica (NEt$_3$) with PE/EtOAc as eluent afforded the target product **3a**.

### General procedure B for the three-component dearomative spirocyclization/imination of nonactivated arenes with non-activated alkenes

A 10 mL vial equipped with a magnetic stir bar was charged with oxime **4a** (0.3 mmol), *N*-benzyl-*N*-(*tert*-butyl)acrylamide **2a** (0.2 mmol), [Ir(dF(CF$_3$)ppy)$_2$(dtbbpy)](PF$_6$) (0.001 mmol) in EtOAc (3 mL). The non-activated alkene **5** (0.4 mmol) was added into the mixture after degassing with N$_2$ for 10 min. Then the mixture was irradiated by blue LEDs ($\lambda$ = 440 nm, 10 W) for 4 h at room temperature. After irradiation, the resulting homogenous solution was transferred to a 25 mL round bottom flask with the aid of CH$_2$Cl$_2$ (2 × 3 mL). NEt$_3$ (approx. 0.5 mL) and SiO$_2$ were added to this solution, and the volatiles were removed under reduced pressure, affording a powder which was loaded on the column. Purification by column chromatography using pre-basified silica (NEt$_3$) with PE/EtOAc as eluent afforded the target product **6a**.

### General procedure C for the three-component dearomative spirocyclization/imination of nonactivated arenes with non-activated alkanes

A 10 mL vial equipped with a magnetic stir bar was charged with oxime **4a** (0.4 mmol), *N*-benzyl-*N*-(*tert*-butyl)acrylamide **2a** (0.2 mmol), cyclohexane **7a** (2 mL), [Ir(dF(CF$_3$)ppy)$_2$(dtbbpy)](PF$_6$) (0.001 mmol) in EtOAc (1 mL). After the degassing with N$_2$ for 10 min, the mixture was irradiated by blue LEDs ($\lambda$ = 440 nm, 10 W) for 4 h at room temperature. After irradiation, the resulting homogenous solution was transferred to a 25 mL round bottom flask with the aid of CH$_2$Cl$_2$ (2 × 3 mL). NEt$_3$ (approx. 0.5 mL) and SiO$_2$ were added to this solution, and the volatiles were removed under reduced pressure, affording a powder which was loaded on the column. Purification by column chromatography using pre-basified silica (NEt$_3$) with PE/EtOAc as eluent afforded the target product **8a**.

### General procedure D for the flow reactions

A 100 mL round bottom flask was loaded with the corresponding oxime (6 mmol), *N*-benzyl-*N*-(*tert*-butyl)acrylamide (4 mmol), [Ir(dF(CF$_3$)ppy)$_2$(dtbbpy)](PF$_6$) (0.02 mmol) in EtOAc (60 mL). The mixture was bubbled with a stream of argon for 20 min (for a three-component reaction), methylenecyclohexane (8.0 mmol) was added after the degassing), then pumped to a homemade flow reactor at a rate of 0.05 mL/min upon the irradiation of 10 W LEDs ($\lambda$ = 440 nm). After irradiation, the resulting homogenous solution was transferred to a 250 mL round bottom flask with the aid of CH$_2$Cl$_2$ (2 × 10 mL). NEt$_3$ (approx. 5 mL) and SiO$_2$ were added to this solution, and the volatiles were removed under reduced pressure, affording a powder which was loaded on the column. Purification by column chromatography using pre-basified silica (NEt$_3$) with PE/EtOAc as eluent afforded the target product.

## Data availability

Materials and methods, detailed optimization studies, computational details, experimental procedures, and mechanistic studies are available in the Supplementary Information. Characterization data and copies of processed NMR spectra for all obtained products and further information (FID) is available from the corresponding author upon request. Source data (xyz coordinates, energies and enthalpies) are provided in this paper.

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

## Acknowledgements

Financial support from the Swedish Research Council (grant no. 2023-04482 (P.D.), grant no. 2020-04764 (M.D.K)), Olle Engkvist Foundation (204-0175), Wenner-Gren Foundation (UPD2020-0228, UPD2021-0145), FORMAS (grant no. 2019-01269), the Magnus Bergvall Foundation, and KTH Royal Institute of Technology are gratefully acknowledged. Financial support from the Russian Science Foundation (project no. 21-73-10211) is gratefully acknowledged. The National Supercomputer Center (NSC) in Linköping is acknowledged for providing computational resources.

## Author contributions

C.Z., M.D.K., and P.D. conceptualized and directed the project. C.Z., E.V.S., and A.S. designed, conducted and analyzed the experiments described in this article. All authors contributed to discussing the results and drafting the manuscript.

## Funding

## Competing interests

The authors declare no competing interests.
