## [Transparent Peer Review file · Nature Communications]

Visible Light-Mediated Dearomative Spirocyclization/Imination of Nonactivated Arenes through Energy Transfer Catalysis

Corresponding Author: Professor Peter Dinér

Version 0:

Reviewer comments:

Reviewer #1

(Remarks to the Author)

The manuscript introduces an innovative photocatalytic strategy for dual functionalization of non-activated aromatic hydrocarbons, recognized for its expansive substrate scope. Despite presenting a new synthetic route, the research's current limitations concerning reaction efficiency and the scope demonstrated temper its overall impact. Given the elevated standards and typical scope of "Nature Communications," the manuscript, as is, may not be suitably positioned for publication in this esteemed journal. However, upon meticulous review, several points are identified for enhancement:

Firstly, numerous examples cited exhibit yields beneath 50% alongside less than desirable diastereomeric ratios (dr values). It is vital to improve these outcomes through meticulous optimization of reaction conditions to bolster the practical utility of the method.

Secondly, while the study centers on benzene derivatives, widening the substrate spectrum to incorporate other aromatic systems would markedly enhance the applicability of the methodology.

Thirdly, the manuscript does not thoroughly explore the potential of the method within the context of medicinal chemistry or the synthesis of bioactive molecules. Incorporating such demonstrations would emphasize the method's practical value.

Lastly, including a discourse on the potential applications of the synthesized products, particularly within pharmaceutical or industrial chemistry contexts, would augment the manuscript's significance.

Attending to these aspects meticulously will bolster the scientific contribution of the work and align it more closely with the expectations of "Nature Communications." The reviewers look forward to a revised submission reflecting these enhancements.

Reviewer #2

(Remarks to the Author)

The present work describes the method for the dearomative difunctionalization by radical spirocyclization of benzyl acrylamides with bifunctional oxime esters. In the reaction, an Ir-based photocatalyst mediates an energy transfer leading to the homolysis of the N-O bond of the oxime esters, resulting in the formation of a relatively long-lived iminyl radical and transient carbon radicals through decarboxylation. The nucleophilic carbon radicals add to benzyl acrylamides, followed by spirocyclization and then coupling with iminyl radical species to yield the final product. When oxime carbonates are used, dearomative spirocyclization was achieved by the addition of suitable alkenes in order to solve the polarity mismatch between the carbonate O-radical and the electron-deficient acrylamides. The use of oxime carbonates has also been applied to the HAT system, where simple aliphatic substrates can be used as carbon radical sources. In this study, these three types of dearomative spirocyclization were achieved, allowing the construction of unique three-dimensional structures. Large-scale experiments using a flow system were also demonstrated. It is also shown that the installed iminyl functionality is readily modifiable by hydrolysis and reduction. Mechanistic studies including DFT calculations can support the mechanism. Overall, the manuscript is well written and contains interesting results. However, this reviewer feels that the

present study is more of an extension of previous work and contains limited advances and novelties. The method of generating two distinct radical species from bifunctional oxime derivatives by energy transfer using a photocatalyst has been well established to date. The present study also employs the same method reported by Glorius and others. In addition, the reaction involving a radical addition to benzyl acrylamides followed by dearomative spirocyclization has already been reported by the author (ref. 28). Although the dearomative difunctionalization including radical amination is a novel finding, the transformation is basically the same as in the previous work. The range of acrylamides is relatively narrow. The substituent on the nitrogen atom is limited to the sterically hindered tert-butyl group. This point should be discussed. Although modification of the amino group of the products was demonstrated, further elaboration of the products is desirable.

In conclusion, I do not recommend accepting this work for publication in Nature Communications at this time.

Reviewer #3

(Remarks to the Author)

This manuscript by Dinér et al. describes the development of a dearomative spirocyclization and imination sequence through visible-light-mediated energy transfer catalysis. The transformation manages to overcome limitations in a previous study from the same group, and forms three or four new bonds in one synthetic step. The authors showed that using bifunctional radical precursors with cleavable N–O bonds, a pair of radicals with well-differentiated reactivity profiles can be generated, and several bond formations can therefore be achieved in the desired sequence. Different radical reactivity modes can be achieved by tuning the structure of the radical precursor. I think the main weakness of the study is the limited scope of its radical acceptor and imination agent. Nonetheless, I find the variety of mechanistic pathways interesting and illuminating, and the products with spiro scaffolds important. I think the manuscript should be considered for publication in Nature Communications, given that the following issues are addressed in a revised version:

1. The limitation on the radical acceptor scope (Figure 3) merits some extra discussion. In particular, any radical-stabilizing para substituents on the aromatic ring seems to not work for the reaction. Can the authors comment on what happened in these attempts? Were the spirocyclizations not successful, or did the spirocyclizations generate a radical that was too stable to react further?
2. The only iminyl radical used in this study is with diphenyl substitution. Have the authors tried to vary the iminyl radical structure in any way? Given that the triplet excited state of 1a (44.3 kcal/mol) seems very accessible through energy transfer under these conditions, I expected there to be more room to tweak the iminyl radical structure. Of course, it could be because alternative substitution patterns would generate iminyl radicals that are not persistent enough to allow the desired transformation. I would appreciate the authors' input on this issue.
3. In Figure 2 and the associated discussion, compounds 4a and 5a were not clearly defined. In fact, we as readers are unable to match these compound numbers to their structures until Figure 4. Please consider adding the structures of 4a and 5a to Figure 2 to avoid this confusion.
4. The calculations were conducted with B3LYP-D3. While this should be sufficient for geometry optimizations, I think single-point energy calculations should be conducted with a more accurate functional, such as M06-2X or ω B97X-D, as these functionals are known to perform much better for reaction barriers than B3LYP-D3. The single-point energy obtained from M06-2X or ω B97X-D calculations (a.k.a. the "SCF Done" energy values in Gaussian outputs) can then be added to the Gibbs free energy correction from the B3LYP-D3 optimization/frequency calculation (the "Thermal correction to Gibbs Free Energy" values in Gaussian outputs) to obtain a more accurate Gibbs free energy value. The authors should re-evaluate Gibbs free energy values for all calculated structures in this way.

Reviewer #4

(Remarks to the Author)

In this manuscript, Dinér and their co-workers report a versatile photocatalytic method for rapid building spiro-imine scaffolds from nonactivated arenes via energy transfer catalysis. This method works smoothly for a series of bifunctional oxime esters and carbonates. Besides, the strategy also works well for multi-component reactions. Further, the mechanism of the reaction is proposed based on experimental and theoretical DFT calculation studies. Therefore, I would recommend this article to be published in Nature Communication after minor revisions.

1. Some mechanistic experiments should be added, including radical trapping experiments as well as direct excitation studies.
2. The structure of the paper can be adjusted appropriately, for example, the substrate scope can be written first, and the reaction mechanism or DFT calculations can be presented later for reading.
3. For substrate 2, is it necessary to have a substituent on the nitrogen atom? How about the reactivity?
4. Figure 3, the structure of some compounds should be optimized, such as 3f, 3l, 3r and so on.
5. The structures of some compounds in SI should be optimized, such as 1f, 1h.
6. In SI, the table of contents should be streamlined. The background of Figure S6 should be adjusted.

Version 1:

Reviewer comments:

Reviewer #2

(Remarks to the Author)

The author has carefully considered the reviewers' comments and has made the necessary corrections. Extensive results from additional experiments have been added, which significantly improve the manuscript, particularly in terms of substrate scope. Therefore, I now think that the revised manuscript has reached a level of quality suitable for publication in Nature Communications. However, there are still some comments that should be addressed before acceptance.

1) In the supporting information, NMR spectra should be listed in the order of compound numbers.

2) Additional experiments using acrylamides bearing various aromatic rings have been conducted (3i–3m). Regarding reactions using substrates with unsymmetrical aromatic systems, the author states that there are "8 possible stereoisomers from the three stereogenic centers of the product", but this count might include enantiomers. Since this study focuses only on diastereomers, the author should clarify how many diastereomers each compound can theoretically produce and specify how many were actually obtained. In this regard, product 3m was obtained as a mixture of four diastereomers, whereas 3l was obtained as a mixture of two. A discussion of the expected structures would be helpful.

3) The characterization of products 3i–3m should be conducted more carefully. Some isomers were obtained with certain levels of impurities, as indicated by the larger-than-expected integral values in the aromatic region of the ¹H NMR spectra. These compounds should be further purified and re-characterized.

Reviewer #3

(Remarks to the Author)

The authors have satisfactorily addressed all of my concerns and I recommend the manuscript for publication in Nature Communications.

Reviewer #4

(Remarks to the Author)

I thoroughly looked into the issues brought by reviewer 1. After carefully reviewed the revised manuscript, I believe that the authors have properly addressed these issues, which further strengthens the publication of this work in the journal.

The point-to-point response to the comments from reviewers

Reviewer 1:

Recommendation: Attending to these aspects meticulously will bolster the scientific contribution of the work and align it more closely with the expectations of "Nature Communications." The reviewers look forward to a revised submission reflecting these enhancements.

Q1. Firstly, numerous examples cited exhibit yields beneath 50% alongside less than desirable diastereomeric ratios (*dr* values). It is vital to improve these outcomes through meticulous optimization of reaction conditions to bolster the practical utility of the method.

Response: In the development of our disclosed transformation, a wide array of parameters, including photocatalyst, solvent, irradiation time, ratio of each component, intensity of light source, and temperature were screened to offer moderate yields in the decarboxylation pathway as shown in Table S1-5 (see below or supporting information). Under the optimized conditions, both acrylamide acceptor and the oxime ester were completely consumed after the irradiation. If one examines the suggested catalytic cycle, the "moderate" yield origins from the complexity of the developed reaction that involve 1) N–O bond cleavage, 2) $2 \times$ C–C bond forming events and a radical-radical heterocoupling step (C–N bond formation) with concurrent formation of four different types of radical intermediates and three new chemical bonds. As for the direct radical addition pathway, it's more challenging since four new chemical bonds (C–O, C–C, C–C and C–N bonds) are formed through a radical cascade sequence, including five different types of radical intermediates. This suggest that the average efficiency for the different steps is around 90% ($0.90^5 = 59\%$), which is not that bad considering the complexity and the number of bonds formed in reaction manifold. In addition, replacing the phenyl ring with dibenzofuran or introducing a fluoro-substituent in the 2-position of the aromatic ring led to over 70% yield (**3ad** and **3af**), demonstrating the high efficiency of this dearomatization/imination strategy.

Any diastereomeric ratio (*dr*) that is observed arises in the heterocoupling step between the iminyl radical that adds to either side of the planar dearomatized ring (**IM5**). The energy difference between the two diastereomers (*cis*- and *trans*-**3a**) are negligible (less than 1 kcal mol⁻¹, see Figure 2, bottom) and they are formed in a barrierless process as suggested by DFT. The negligible Gibbs free energy difference and the lack of transition state suggest that the two formed stereocenters are too far apart (C1 and C5) to be able to induce any diastereoselectivity.

Tables added to SI:

Table S1. The optimization for the two-component dearomative spirocyclization/imination of nonactivated arenes.^a

Entry	PC (1 mol%)	Solvent	Yield (%) ^b	dr ^b
1	[Ir(dFCF ₃ ppy) ₂ (dtbbpy)]PF ₆	EtOAc	50	1.4:1
2	Ir(ppy) ₃	EtOAc	30	1.3:1
3	4CzIPN	EtOAc	34	1.4:1
4	3DPAFIPN	EtOAc	5	1.4:1
5	4DPAIPN	EtOAc	<5	1.4:1

6 ^c	Thioxanthen-9-one	EtOAc	38	1.3:1
7 ^{c,d}	Benzophenone	EtOAc	40	1.4:1
8	[Ir(dFCF ₃ ppy) ₂ (dtbbpy)]PF ₆	CH ₃ CN	38	1.3:1
9	[Ir(dFCF ₃ ppy) ₂ (dtbbpy)]PF ₆	CH ₂ Cl ₂	44	1.3:1
10	[Ir(dFCF ₃ ppy) ₂ (dtbbpy)]PF ₆	PhCF ₃	46	1.4:1
11	[Ir(dFCF ₃ ppy) ₂ (dtbbpy)]PF ₆	acetone	47	1.4:1
12	[Ir(dFCF ₃ ppy) ₂ (dtbbpy)]PF ₆	DMSO	34	1.4:1
13	[Ir(dFCF ₃ ppy) ₂ (dtbbpy)]PF ₆	DMF	44	1.3:1
14 ^e	[Ir(dFCF ₃ ppy) ₂ (dtbbpy)]PF ₆	EtOAc	45	1.4:1
15 ^f	[Ir(dFCF ₃ ppy) ₂ (dtbbpy)]PF ₆	EtOAc	46	1.4:1
16 ^g	[Ir(dFCF ₃ ppy) ₂ (dtbbpy)]PF ₆	EtOAc	54	1.4:1
17 ^{g,h}	[Ir(dFCF ₃ ppy) ₂ (dtbbpy)]PF ₆	EtOAc	56	1.4:1
18	--	EtOAc	0	--
19 ⁱ	[Ir(dFCF ₃ ppy) ₂ (dtbbpy)]PF ₆	EtOAc	0	--
20 ^d	- (390 nm)	EtOAc	25	1.4:1

^a Reaction conditions: **1a** (0.3 mmol, 1.5 equiv.), **2a** (0.2 mmol, 1.0 equiv.), photocatalyst (1 mol%), solvent (3 mL), N₂, blue LEDs (440 nm), 2 h, room temperature. ^b Yields and dr were determined by ¹H NMR using 4-nitrobenzotrile as the internal standard. ^c 20 mol% photocatalyst. ^d Violet LEDs (390 nm). ^e 1.5 mL EtOAc). ^f 6.0 mL EtOAc). ^g 0.5 mol% photocatalyst. ^h 10 w light intensity. ⁱ no light.

Table S2. The optimization of the loading of photocatalyst and reaction time.

Entry	[Ir(dFCF ₃ ppy) ₂ (dtbbpy)]PF ₆	Reaction time	Yield of 3a (%)	dr	Conversion 1a (%)	Conversion 2a (%)
1	0.5 mol%	4 h	54	1.4:1	100	>95
2	0.25 mol%	4 h	56	1.4:1	100	>95
3	0.125 mol%	4 h	42	1.4:1	54	81
4	0.25 mol%	1 h	47	1.4:1	67	91
5	0.25 mol%	2 h	53	1.4:1	77	93

Reaction conditions: **1a** (0.3 mmol, 1.5 equiv.), **2a** (0.2 mmol, 1.0 equiv.), [Ir(dFCF₃ppy)₂(dtbbpy)]PF₆, EtOAc (3 mL), N₂, blue LEDs (440 nm), room temperature. ^b Yields and dr were determined by ¹H NMR using 4-nitrobenzotrile as the internal standard.

Table S3. The optimization of the ratio of **1a** and **2a**.

Entry	1a (mmol)	2a (mmol)	Yield of 3a (%)	dr	Conversion of 1a (%)	Conversion of 2a (%)
1	0.3	0.2	53	1.4:1	77	93
2	0.2	0.2	42	1.4:1	94	74
3	0.2	0.3	46	1.4:1	100	44
4	0.2	0.4	44	1.4:1	100	28

Reaction conditions: **1a**, **2a**, [Ir(dFCF₃ppy)₂(dtbbpy)]PF₆ (0.25 mol%), EtOAc (3 mL), N₂, blue LEDs (440 nm), room temperature, 2 h. Yields and dr were determined by ¹H NMR using 4-nitrobenzonitrile as the internal standard.

Table S4. The optimization of additives.

Entry	Additives (equiv.)	Reaction time	Yield of 3a (%)	dr	Conversion 1a (%)	Conversion 2a (%)
1	NBu ₄ OPO(OtBu) ₂ (0.2)	1 h	49	1.4:1	73	81
2	Collidine (1.0)	1 h	49	1.4:1	100	91
3	NaOAc (1.0)	1 h	50	1.4:1	100	95
4	Sc(OTf) ₃ (0.2)	2 h	50	1.4:1	86	>95
5	CuBr ₂ (0.2)	2 h	11	1.4:1	42	35

Reaction conditions: **1a** (0.3 mmol, 1.5 equiv.), **2a** (0.2 mmol, 1.0 equiv.), [Ir(dFCF₃ppy)₂(dtbbpy)]PF₆ (1 mol%), EtOAc (3 mL), N₂, blue LEDs (440 nm), room temperature. Yields and dr were determined by ¹H NMR using 4-nitrobenzonitrile as the internal standard.

Table S5. The optimization of light intensity.

Entry	Light intensity	Reaction time	Yield 3a (%)	dr	Conversion 1a (%)	Conversion 2a (%)
1	10 w	2 h	56	1.4:1	77	>95
2	20 w	2 h	56	1.4:1	78	>95
3	30 w	2 h	53	1.4:1	83	>95
4	40 w	2 h	51	1.4:1	84	>95

Reaction conditions: **1a** (0.3 mmol, 1.5 equiv.), **2a** (0.2 mmol, 1.0 equiv.), Ir(dFCF₃ppy)₂(dtbbpy)]PF₆ (0.5 mol%), EtOAc (3 mL), N₂, blue LEDs (440 nm), room temperature. Yields and dr were determined by ¹H NMR using 4-nitrobenzonitrile as the internal standard.

Table S6. The optimization of flow condition

Entry	Light intensity	Reaction time	Temperature	Yield 3a (%)	dr	Conversion 1a (%)	Conversion 2a (%)
1	40 w	10 min	r.t.	37	1.4:1	54	69
2	40 w	20 min	r.t.	49	1.4:1	78	79
3	40 w	2 h	r.t.	54	1.4:1	100	>95
4	10 w	2 h	r.t.	54	1.4:1	100	>95
5	10 w	4 h	5 °C	59	1.4:1	100	>95
6	10 w	12 h	-10°C	62	1.4:1	100	>95

Reaction conditions: **1a** (0.3 mmol, 1.5 equiv.), **2a** (0.2 mmol, 1.0 equiv.), $[\text{Ir}(\text{dFCF}_3\text{ppy})_2(\text{dtbbpy})]\text{PF}_6$ (1 mol%), EtOAc (3 mL), N_2 , blue LEDs (440 nm). Yields and dr were determined by ^1H NMR using 4-nitrobenzonitrile as the internal standard.

Q. 2. Secondly, while the study centers on benzene derivatives, widening the substrate spectrum to incorporate other aromatic systems would markedly enhance the applicability of the methodology.

Response: Several additional benzene derivatives have been included in the substrate scope. This includes several unsymmetrical substituted systems, such as thiophene (**3ac**), benzofuran (**3ad**), *para*-substituted aromatic systems (**3ae** and **3ag**) and a 2-fluorosubstituted system (**3af**). These additions demonstrate that heteroatoms are indeed tolerated in the aromatic scaffolds and the substituents are tolerated in the *para*-position, leading to highly substituted products. Furthermore, in addition to acrylamides, a BCB-tethered benzamide has been found to be well-tolerated to construct the unique tricycle with two spiro centers in one-step reaction (**3ah**), which significantly expands the scope compared to previous iterations.

Additional aromatic systems included in the substrate scope:

Q. 3. Thirdly, the manuscript does not thoroughly explore the potential of the method within the context of medicinal chemistry or the synthesis of bioactive molecules. Incorporating such demonstrations would emphasize the method's practical value. **Q. 4.** Lastly, including a discourse on the potential applications of the synthesized products, particularly within pharmaceutical or industrial chemistry contexts, would augment the manuscript's significance.

Response: To show the potential of the protocol within the context of medicinal chemistry or bioactive molecules, several drug substrates and biomolecules (amino acid and sugar-based substrates) were used as substrates. We now demonstrate that 6 additional pharmaceutical compounds and bioactive compounds, such as the amino acid derivative tosylglycine and the sugar-based diprogolic acid. These new examples highlight that the developed protocol can be applied in late-stage functionalization of pharmaceuticals compounds.

Expanded pharmaceutical compounds in the substrate scope:

Reviewer 2:

Recommendation: In conclusion, I do not recommend accepting this work for publication in Nature Communications **at this time.**

Q. 1. The method of generating two distinct radical species from bifunctional oxime derivatives by energy transfer using a photocatalyst has been well established to date. The present study also employs the same method reported by Glorius and others.

Response: While bifunctional oxime derivatives were indeed employed in this study, the primary objective

was to achieve the dearomative difunctionalization of nonactivated arenes. Beyond the utilization of iminyl *N*-radicals from oxime derivatives, we are also exploring the potential of other persistent radicals to participate in this transformation, highlighting its potential in other dearomative difunctionalization manifolds. The direct dearomative imination to introduce primary amine equivalents is unprecedented in literature and will open new avenues to functionalize aromatic scaffolds. In addition to the known generation of two distinct radical species from bifunctional oxime derivatives, we have established several ways (via Giese-type coupling, cascade type activation and hydrogen atom transfer) to show that our imination is a versatile and is far from trivial in this context. Our expanded substrate scope clearly indicates that the developed protocol is not just an expansion of documented protocols but rather a completely new methodology for introducing primary amine equivalents via spirocyclic dearomatization.

Expanded substrate scope of the reaction:

Optimized reaction conditions for decarboxylative spirocyclization

Aliphatic carboxylic acid substrates

Keto acids and substituted benzyl acrylamides

Pharmaceuticals

Amino acid and sugar

Intramolecular spirocyclization

Q. 2. In addition, the reaction involving a radical addition to benzyl acrylamides followed by dearomative spirocyclization has already been reported by the author (ref. 28). Although the dearomative difunctionalization including radical amination is a novel finding, the transformation is basically the same as in the previous work. The range of acrylamides is relatively narrow.

Response: The current transformation represents a significant departure from our previous work and should not be regarded as a mere extension thereof. Initially, we encountered challenges with the reactivity of the cyclohexadienyl C-radical intermediate, which exhibited a propensity for both one-electron reduction and oxidation. Despite exhaustive exploration involving various substrates, such as olefins, alkynes, heteroareamics, Bpin₂, and even transition metals, we were unable to effectively trap the cyclohexadienyl C-radical intermediate to facilitate the formation of new C–C or C–heteroatom bonds. The dearomative difunctionalization of nonactivated arenes proved to be particularly challenging and underexplored in this context. In our current approach, we have addressed these challenges by employing persistent N-centered iminyl radicals derived from oxime-based bifunctional derivatives for efficient coupling with the cyclohexadienyl C-radical, leveraging the principles of the persistent radical effect. Furthermore, in addition to acrylamides, BCB-tethered benzylamides were found to be well-tolerated in this reaction, significantly expanding its scope compared to previous iterations. Also, several additional benzene derivatives have been included into the substrate scope. This includes several unsymmetrical substituted systems, such as thiophene (**3ac**), benzofuran (**3ad**), *para*-substituted aromatic system (**3ae** and **3ag**) and a 2-fluorosubstituted system (**3af**). These substrates demonstrate that heterocycles are tolerated and that substituents are tolerated in the *para*-position, leading to highly decorated products.

Expanded substitution pattern in aromatic system:

Except the acrylamide acceptors, BCB-tethered benzylamide was shown to be compatible under the developed conditions yielding the unique tricyclic compound with two spiro centers in one-step (**3ah**, see modified Figure 5 below and in manuscript).

Text added to manuscript:

Recently, strain-release transformations of bicyclo[1.1.0]butanes (BCB) have facilitated the formation of reactive radical intermediates via photocatalysis to yield molecular frameworks in organic synthesis through the release of substantial ring strain energy.⁵⁶⁻⁶⁴

To further show the synthetic versatility of our developed methodology, we sought to utilize our *N*-tert-butyl-*N*-benzyl motif to access bis-spirocyclic products via a radical cascade process involving energy-transfer, strain-release activation, spirocyclization and C–N bond forming iminium.

Modified figure in manuscript:

Figure 5. Synthetic applications for dearomative spirocyclic/imination cascades. *Top:* BCB-enabled formation of *bis*-spirocycles. *Middle:* Large-scale decarboxylative dearomative difunctionalization under continuous flow conditions. *Bottom left:* Deprotection of the imine functionality for selected products. *Bottom right:* Reduction of the imine functionality.

Q. 3. The substituent on the nitrogen atom is limited to the sterically hindered *tert*-butyl group. This point should be discussed.

Response: Free acrylamide was examined and unfortunately did not work for this transformation. The *t*-Bu substituent on the nitrogen atom is essential and decreases the spatial distance between the two carbons that form the spirocyclic center (*ACS Catal.* **2021**, 11, 4968–4972). However, it has been shown that the *tert*-butyl group can be easily removed (*Tetrahedron*, **2014**, 41, 7593) under mild conditions using copper(II) triflate. Additional text has been added to the manuscript as suggested by the reviewer.

Additional text in manuscript:

Smaller modifications of the amide functionality in the somophile, such as substitution of the methylene group with a carbonyl functionality, provided the corresponding spiro-succinimide product (**3q**, 43%, 3:1 dr) in similar yields. Substitution of the *N*-*t*Bu group with *N*-*i*Pr led to a decrease in yield (**3r**, 29%) while having secondary amide gave no product formation. Previously, the importance of the *t*-Bu substituent on the amide functionality was attributed to the decreased spatial distance between the two carbons that undergo the spirocyclization,³⁰ but it has been shown that the *tert*-butyl group can be easily removed under mild conditions using copper(II) triflate.⁴⁷

Q. 4. Although modification of the amino group of the products was demonstrated, further elaboration of the products is desirable.

Response: In this context, the diphenyliminyl group is a primary amine equivalent and the reduction and its deprotection have been shown in the paper.

Reviewer 3:

Recommendation: I think the manuscript *should be considered for publication in Nature Communications*, given that the following issues are addressed in a revised version.

Q. 1. The limitation on the radical acceptor scope (Figure 3) merits some extra discussion. In particular, any radical-stabilizing *para* substituents on the aromatic ring seems to not work for the reaction. Can the authors comment on what happened in these attempts? Were the spirocyclizations not successful, or did the spirocyclizations generate a radical that was too stable to react further?

Response:

Several new benzene derivatives have been included into the substrate scope. This includes several unsymmetrical substituted systems such as thiophene (**3i**), benzofuran (**3j**), *para*-substituted aromatic system (**3l** and **3m**) and a 2-fluorosubstituted system (**3k**). These substrates demonstrates that heteroatoms are tolerated in the aromatic scaffolds and that substituents are tolerated in the *para*-position leading to highly substituted products.

Additional aromatic system included in the substrate scope:

Q. 2. The only iminyl radical used in this study is with diphenyl substitution. Have the authors tried to vary the iminyl radical structure in any way? Given that the triplet excited state of **1a** (44.3 kcal/mol) seems very accessible through energy transfer under these conditions, I expected there to be more room to tweak the iminyl radical structure. Of course, it could be because alternative substitution patterns would generate iminyl radicals that are not persistent enough to allow the desired transformation. I would appreciate the authors' input on this issue.

Response: The reactivity of differently substituted oxime esters has previously been investigated by Glorius and co-workers (*Angew. Chem. Int. Ed.* **2020**, *59*, 3172). Their work demonstrated that the parent oxime from benzophenone (RCO₂NPh₂) are superior to other substitution patterns for the diradical formation and addition to electron-deficient alkenes.

Q. 3. In Figure 2 and the associated discussion, compounds **4a** and **5a** were not clearly defined. In fact, we as readers are unable to match these compound numbers to their structures until Figure 4. Please consider adding the structures of **4a** and **5a** to Figure 2 to avoid this confusion.

Response: We agree that the current labeling was not so clear. For clarity, we have simplified the graph with the fluorescent quenching to include only the structures that are drawn out in Figure 2. In addition, we have added the structures in each graph in the supporting information to increase the clarity for the reader.

Additionally, we have added a paragraph with a reference to the supporting information to more clearly explain the fluorescent quenching for **4a** and **5a**.

Added text to manuscript:

The fluorescence quenching studies for this catalytic system show that only the oxime carbonate **4a** effectively quenches the excited iridium photocatalyst (Figure 2, *top*), while no appreciable quenching was observed for the electron-rich alkene **5a** (see Figure S3, S4 and S5).

Q. 4. *The calculations were conducted with B3LYP-D3. While this should be sufficient for geometry optimizations, I think single-point energy calculations should be conducted with a more accurate functional, such as M06-2X or ω B97X-D, as these functionals are known to perform much better for reaction barriers than B3LYP-D3. The single-point energy obtained from M06-2X or ω B97X-D calculations (a.k.a. the “SCF Done” energy values in Gaussian outputs) can then be added to the Gibbs free energy correction from the B3LYP-D3 optimization/frequency calculation (the “Thermal correction to Gibbs Free Energy” values in Gaussian outputs) to obtain a more accurate Gibbs free energy value. The authors should re-evaluate Gibbs free energy values for all calculated structures in this way.*

Response: Single-point energy calculations have been conducted with the M06-2X functional and was added to the supporting information of the paper. An additional paragraph was added to the computational details of the supporting and Figure S9 has been adjusted with the new results.

Text added to supporting information:

To examine the validity of the selected computational method, the structures obtained at the B3LYP-level of theory were subjected to a single-point calculation using M062X/6-311+G(d,p) and M06/6-311+G(d,p) with the Conductor-like Polarizable Continuum Model (CPCM, Ethylacetate) using the parameters for ethylacetate and the default Unified Force Field radii (UFF). The Gibbs free energies were obtained by adding the thermal correction to Gibbs Free Energy from the B3LYP level of theory. The comparison between relative free energies calculated with different functionals show that the major difference in the formation of the aliphatic and iminyl radicals is seen in the excitation energy from the ground state to the triplet state of the iridium photocatalyst. Both the B3LYP and the M06 functionals reproduce the experimental triplet energy (58 kcal mol⁻¹ compared to 61.1 kcal mol⁻¹) while the M062X strongly overestimates excitation energy of the triplet state of the iridium photo catalyst. However, with the exception for the iridium photocatalyst excitation, both B3LYP, M062X and M06 functionals show a similar potential energy surface from the triplet **1a*** (see Figure S9A). In addition, the structures obtained at the B3LYP-level of theory for the dearomative spirocyclization/imination were subjected to a single-point calculation using M062X/6-311+G(d,p) with the Conductor-like Polarizable Continuum Model (CPCM, Ethylacetate) using the parameters for ethylacetate and the default Unified Force Field radii (UFF). The results show similar relative Gibbs free energy and barrier for the dearomative spirocyclization/imination for the two DFT methods (Figure S9B).

Figures added to supporting information:

Gibbs free energy diagrams

A) Activation via energy transfer catalysis

B) Gibbs free energy surface for dearomative spirocyclization/imination

Figure S9. Gibbs free energy surface (red) for the formation of C- and N-centered radicals optimized at the B3LYP/6-311+G(d,p) level using the Grimme correction for dispersion (D3) and the Conductor-like Polarizable Continuum Model (CPCM, UFF, Ethylacetate). Gibbs free energies are given in kcal mol⁻¹. Values in blue and black refer to single-point energy calculations at the M062X or M06/6-311+G(d,p) level using the Conductor-like Polarizable Continuum Model (CPCM, UFF, Ethylacetate) with the addition of Gibbs free energy-corrections from the B3LYP level of theory. Gibbs free energies are given in kcal mol⁻¹.

Reviewer 4:

Recommendation: I would recommend this article to be published in Nature Communications after minor revisions.

Q. 1. Some mechanistic experiments should be added, including radical trapping experiments as well as direct excitation studies.

Response: The direct excitation using blue LEDs (440 nm) did not lead to product formation (Table S1, entry 19). This shows that the presence of the iridium photocatalyst is important for an efficient process using visible light. However, the direct excitation without photocatalyst but with shorter wavelength (390 nm) led to the formation of target product **3a** with 25% (Table S1, entry 20). This clearly shows that the photocatalysed reaction is more efficient than direct excitation and avoids the use of irradiation sources of shorter wavelengths.

Several different trapping experiments and analysis of product mixtures were performed to gain insight into mechanistic pathways of the reaction. From the analysis of the reaction mixture, the O-alkylated oxime (**A1**) indicate that both the aliphatic radical ($\text{PhCH}_2\text{CH}_2\cdot$) and iminyl radical ($\cdot\text{N}=\text{Ph}_2$) are formed as intermediates in the EnT process (see Figure S6). Additionally, we also observed the homocoupled products from the iminyl radicals (**A2**) ($\cdot\text{N}=\text{Ph}_2$) and dearomatized radicals (**A3**), which is typical for more persistent radicals (see Figure S6). The addition of TEMPO (2 equiv.) to the reaction mixture under otherwise standard conditions did not lead to product formation and instead the formation of the alkylated TEMPO was detected by HRMS. An additional text about this experiment has been added to the manuscript.

In the case of the three-component system utilizing the carbonate oxime radical precursor, the product analysis show that the main product originates from carbonate radical addition to the electron-rich alkene, followed by a second addition of the formed C-centered radical to the more electron-poor alkene. In the absence of the electron-rich alkene the target product was not observed (Figure S8, *top*). In the absence of the alkene electron-poor alkene the oxyimination product was detected as reported previously by Glorius and Molander (Figure S8, *middle*). Finally, in the absence of any somophiles, we detect the product for the direct amination of the cyclohexane solvent (Figure S8, *bottom*), which occurs via hydrogen atom transfer between the carbonate radical and the solvent.

Modified text in manuscript:

These side-products include the radical-radical coupling adducts between the proposed intermediates, including one C–N cross-coupled between the iminyl radical the aliphatic radical and two homodimerization adducts (Figure 2, *top right*; Figure S6).

Additional trapping experiments in the presence of the radical scavenger TEMPO (2 equiv) during otherwise standard conditions completely inhibited the formation of product **3a** and led to the formation of the coupling product between TEMPO and the C-centered aliphatic radical as detected by HRMS (Figure S6).

Figures in supporting information:

Detection of side-products using HRMS

Trapping experiment using TEMPO

Figure S6. LC-MS spectra for the two-component dearomative spirocyclization/imination of nonactivated arenes.

Additional control experiments

Figure S8. Control experiments for reactions with carbonate oximes.

Modified text in manuscript:

Hence, unactivated alkanes can be exploited as the source to form C-centered radicals that can engage in numerous reactions with somophiles. The versatility of the carbonate radical was tested in control experiments. In the absence of an electron-rich alkene, no spirocyclic-imation product was formed suggesting a competing reactivity for the carbonate radical (Figure S8a). On the other hand, performing a similar reaction in the absence of an electron-poor alkene and in the presence of an electron-rich alkene yields the difunctionalized product (Figure S8b) as previously reported by Molander and Glorius.³⁶⁻³⁹ Upon irradiation of 4a, the imination product of cyclohexane was detected by HRMS which shows the potential HAT capabilities of the carbonate radical.

Light on-off experiment

Figure S7. Light on-off experiment. General procedure: eight 10 mL Pyrex tubes equipped with a magnetic stir bar were separately charged with oxime **1a** (0.3 mmol), *N*-benzyl-*N*-(*tert*-butyl)acrylamide **2a** (0.2 mmol), $[\text{Ir}(\text{dFCF}_3\text{ppy})_2(\text{dtbbpy})]\text{PF}_6$ (0.001 mmol) in EtOAc (3 mL). After the degassing with N_2 for 10 mins, the reactions were placed in light and dark in every alternative 5 minutes. Then the resulting homogenous solutions were transferred to a 25 mL round bottom flask. Yields were determined by ^1H NMR using 4-nitrobenzonitrile as the internal standard.

Q. 2. The structure of the paper can be adjusted appropriately, for example, the substrate scope can be written first, and the reaction mechanism or DFT calculations can be presented later for reading.

Response: The authors considered to change the layout of the paper but have decided to keep the original structure of the paper.

Q. 3. For substrate 2, is it necessary to have a substituent on the nitrogen atom? How about the reactivity?

Response: Acrylamides housing an N-H moiety were examined and did not work for this transformation. The *t*-Bu substituent on the nitrogen atom is essential and decreases the spatial distance between the two carbons

that form the spirocyclic center (*ACS Catal.* **2021**, 11, 4968–4972). However, it has been shown that the *tert*-butyl group can be easily removed (*Tetrahedron*, **2014**, 41, 7593) under mild conditions using copper(II) triflate. Additional text has been added to the manuscript as suggested by the reviewer.

Additional text in manuscript:

Smaller modifications of the amide functionality in the somophile, such as substitution of the methylene group with a carbonyl functionality, provided the corresponding spiro-succinimide product (**3q**, 43%, 3:1 dr) in similar yields. Substitution of the *N*-*t*Bu group with *N*-*i*Pr led to a decrease in yield (**3r**, 29%) while having secondary amide gave no product formation. Previously, the importance of the *t*-Bu substituent on the amide functionality was attributed to the decreased spatial distance between the two carbons that undergo the spirocyclization,³⁰ but it has been shown that the *tert*-butyl group can be easily removed under mild conditions using copper(II) triflate.⁴⁷

Q. 4. *Figure 3, the structure of some compounds should be optimized, such as 3f, 3l, 3r and so on.*

Response: Thank you very much for your suggestion. The structure of **3f**, **3l** and **3r** in Figure 3 has been adjusted in the revised manuscript.

Q. 5. *The structures of some compounds in SI should be optimized, such as 1f, 1h.*

Response: The structure of **1f** and **1h** in SI has been adjusted in the revised manuscript.

Q. 6. *In SI, the table of contents should be streamlined. The background of Figure S6 should be adjusted.*

Response: The Table of content was streamlined in the SI and Figure 6 has been adjusted in the revised SI.

Response to Reviewers comments:

The author has carefully considered the reviewers' comments and has made the necessary corrections. Extensive results from additional experiments have been added, which significantly improve the manuscript, particularly in terms of substrate scope. Therefore, I now think that the revised manuscript has reached a level of quality suitable for publication in Nature Communications. However, there are still some comments that should be addressed before acceptance.

1) *In the supporting information, NMR spectra should be listed in the order of compound numbers.*

Response: The NMR spectra have been listed in the order of compound numbers.

2) Additional experiments using acrylamides bearing various aromatic rings have been conducted (3i–3m). Regarding reactions using substrates with unsymmetrical aromatic systems, the author states that there are "8 possible stereoisomers from the three stereogenic centers of the product", but this count might include enantiomers. Since this study focuses only on diastereomers, the author should clarify how many diastereomers each compound can theoretically produce and specify how many were actually obtained. In this regard, product 3m was obtained as a mixture of four diastereomers, whereas 3l was obtained as a mixture of two. A discussion of the expected structures would be helpful.

Response: We agree that the use the term "8 possible stereoisomers from the three stereogenic centers of the product" is a bit misleading for the audience reader in this context. We have added a paragraph in the manuscript clarifying that.

Changed paragraph in the manuscript:

"Next, we wanted to investigate the spiroiminative reactivity of the aromatic moiety. It should be noted that the obtained compounds from the unsymmetrically substituted aromatic systems have 4 possible diastereomeric pairs arising from the three stereogenic centers of the product, and thus, the target compounds were either isolated as individual compounds or as a mixture of stereoisomers (see Figure 3)."

3) The characterization of products 3i–3m should be conducted more carefully. Some isomers were obtained with certain levels of impurities, as indicated by the larger-than-expected integral values in the aromatic region of the ¹H NMR spectra. These compounds should be further purified and re-characterized.

Response: We have revisited the spectra carefully and after careful examination we agree that for example compound **3m** contains degrees of impurities. After several attempts to get a purer sample, we decided to leave compound **3m** out of the manuscript. The compounds have been renamed in the manuscript and supporting information to accommodate the changes made in compound numbering.

Best regards

Peter Dinér